# Survival of *Escherichia coli* in Airborne and Settled Poultry Litter Particles

**DOI:** 10.3390/ani12030284

**Published:** 2022-01-24

**Authors:** Xuan Dung Nguyen, Yang Zhao, Jeffrey D. Evans, Jun Lin, Joseph L. Purswell

**Affiliations:** 1Department of Animal Science, The University of Tennessee, Knoxville, TN 37996, USA; ndung@vols.utk.edu (X.D.N.); jlin6@utk.edu (J.L.); 2Poultry Research Unit, Agriculture Research Service, United States Department of Agriculture (USDA), Mississippi State, MS 39762, USA; jeff.evans@usda.gov (J.D.E.); joseph.purswell@usda.gov (J.L.P.)

**Keywords:** airborne *E. coli*, settled *E. coli*, survivability, airborne transmission, poultry

## Abstract

**Simple Summary:**

Airborne transmission is recognized as an important mechanism of disease spreading in livestock and poultry production, yet is far from being fully understood. Evaluating the impact of airborne transmission requires information of the microbial survivability. We determined the survivability of the *E. coli*—a common microbial species found in poultry environment—in airborne particles, settled dust, and poultry litter under laboratory environmental conditions. The poultry litter which contained mainly manure mixed with fresh wood shavings was collected from a commercial farm. Results of the study showed that the half-life time of airborne *E. coli* was 5.7 ± 1.2 min. The half-life time of *E. coli* in poultry litter and settled particles was 15.9 ± 1.3 h and 9.6 ± 1.6 h, respectively. The findings of this study will help better estimate the impact of airborne transmission of *E. coli* in poultry production.

**Abstract:**

Airborne *Escherichia coli* (*E. coli*) in the poultry environment can migrate inside and outside houses through air movement. The airborne *E. coli*, after settling on surfaces, could be re-aerosolized or picked up by vectors (e.g., caretakers, rodents, transport trucks) for further transmission. To assess the impacts of airborne *E. coli* transmission among poultry farms, understanding the survivability of the bacteria is necessary. The objective of this study is to determine the survivability of airborne *E. coli*, settled *E. coli*, and *E. coli* in poultry litter under laboratory environmental conditions (22–28 °C with relative humidity of 54–63%). To determine the survivability of airborne *E. coli*, an AGI-30 bioaerosol sampler (AGI-30) was used to collect the *E. coli* at 0 and 20 min after the aerosolization. The half-life time of airborne *E. coli* was then determined by comparing the number of colony-forming units (CFUs) of the two samplings. To determine the survivability of settled *E. coli*, four sterile Petri dishes were placed on the chamber floor right after the aerosolization to collect settled *E. coli*. The Petri dishes were then divided into two groups, with each group being quantified for culturable *E. coli* concentrations and dust particle weight at 24-h intervals. The survivability of settled *E. coli* was then determined by comparing the number of viable *E. coli* per milligram settled dust collected in the Petri dishes in the two groups. The survivability of *E. coli* in the poultry litter sample (for aerosolization) was also determined. Results show that the half-life time of airborne *E. coli* was 5.7 ± 1.2 min. The survivability of *E. coli* in poultry litter and settled *E. coli* were much longer with the half-life time of 15.9 ± 1.3 h and 9.6 ± 1.6 h, respectively. In addition, the size distribution of airborne *E. coli* attached to dust particles and the size distribution of airborne dust particles were measured by using an Andersen impactor and a dust concentration monitor (DustTrak). Results show that most airborne *E. coli* (98.89% of total *E. coli*) were carried by the dust particles with aerodynamic diameter larger than 2.1 µm. The findings of this study may help better understand the fate of *E. coli* transmitted through the air and settled on surfaces and evaluate the impact of airborne transmission in poultry production.

## 1. Introduction

The United States of America (USA) is one of the leading countries in poultry production. Poultry products originating in the USA primarily consist of meat from broilers and turkeys and eggs from layers. According to the USDA report [1], the combined value of production from these products in 2020 exceeded USD 35 billion. These products provide important and affordable sources of dietary protein to the domestic population. In addition, approximately 18% of the USA poultry products are exported and poultry production in the USA was estimated to provide over 1 million jobs. However, the outbreak of infectious diseases is one of the biggest challenges for the poultry industry. For example, the Highly Pathogenic Avian Influenza (HPAI) outbreaks in the USA in 2015 resulted in losses of over 50 million birds and 3.3 billion dollars [2].

*Escherichia coli* (*E. coli*) is a member of the *Enterobacteriaceae* family and is commonly associated with the intestinal tract of warm-blooded animals and the environment in which these animals reside. In poultry, *E. coli* primarily inhabits the lower gastrointestinal tract as an indicator for the poultry environmental quality and exists there as an important commensal species. Typically, *E. coli* are harmless, but some *E. coli* strains may be pathogenic in nature and their virulence may lead to losses in the poultry industry. Pathogenic *E. coli* strains in poultry are commonly referred to as avian pathogenic *E. coli* (APEC) [3]. The APEC causes the systemic disease colibacillosis in broilers. The severity of APEC disease depends on the health status of the host, virulence characteristics of the *E. coli* strain, and other predisposing factors such as stress. Approximately 30% of broiler flocks in the U.S are infected by subclinical colibacillosis [4].

*E. coli* can be abundant in poultry house with concentrations up to 4 log_10_ CFU m^−3^ in the air [5], 3 log_10_ CFU g^−1^ in feeds [6], and 7 log_10_ CFU g^−1^ in poultry litter [7]. To reduce the economic losses caused by *E. coli*, antibiotics, such as tetracyclines and trimethoprim sulfamethoxazole, have been widely utilized in poultry feed [8]. However, the widespread use of antibiotics can cause the emergence and re-emergence of antibiotic resistant bacterial strains. Thus, the use of antibiotics has been limited and many bacteria, including *E. coli*, have reemerged as significant threats to poultry production. Some alternatives were developed to reduce *E. coli* contamination of the farm microclimate such as probiotics [9] and UV lights [10]. These methods do not rely on the use of antibiotics and are relatively effective in reducing microbial contamination in poultry houses. However, these studies have not mentioned the effectiveness of reducing airborne bacteria which attach to dust particles. Therefore, further studies on airborne *E. coli* attached to dust particles such as their survivability or size distribution which directly affects the effectiveness of the methods are needed to investigate.

The litter is a major reservoir of microorganisms in the poultry environment [11]. The dry matter contents can be about 70–80% of litter mass and it can contain abundant biological organisms and compounds that can affect the quality of the poultry environment [12]. Dust particles are aerosolized because of bird activity, as such, the poultry environment is highly dusty.

Air in the poultry houses may contain abundant microorganisms such as *E. coli* [13]. *E. coli* from manure first deposit into poultry litter and are then aerosolized through bird activities [14]. Ventilation systems can drive their migration across a poultry house or even from barn to barn. Airborne *E. coli* were shown to account for 2–6% of the total airborne bacteria in poultry houses [5]. With the high concentration of *E. coli* and the possibility of barn-to-barn transmission, the airborne *E. coli* can harm the entire wide range of environment outside the poultry houses, and they can deposit on surfaces near the poultry houses. The barn-to-barn airborne transmission of avian influenza was investigated in a study conducted in 2019 [15]. The probability of airborne infection is affected by several factors including farm type, flock size, and distance of transmission where the survivability of the pathogen is among the key factors for the modeling accuracy. Moreover, the survivability of *E. coli* on stainless steel under refrigeration conditions and room temperature was reported to exceed 28 days [16]. Therefore, it is also possible that *E. coli* can persist for a long time on various surfaces in the poultry production environment. With such a long survival period on the surface, they can spread to larger areas through vectors. These all raise the question of how long the airborne *E. coli*, carried by poultry litter particles, can survive in the air and on the physical surfaces when settled.

To determine the survivability of airborne and settled *E. coli* in laboratory, a proper aerosolization method that may mimic the fate of *E. coli* in the commercial poultry production environment is required. The wet aerosolization method such as nebulization was widely used to study the survivability of airborne *E. coli* [17]. However, the airborne *E. coli* in poultry houses are aerosolized from dried litter by bird activities, such as dust bathing [14]. So, the results of the study based on wet aerosolization cannot apply to the actual situation in the poultry house. In addition, the survivability of settled *E. coli* after going through the dry aerosolization process has never been investigated. Therefore, a study to determine the survivability of airborne *E. coli* and settled *E. coli* after being aerosolized based on dry aerosolization method needs to be done.

Size distribution of airborne *E. coli* attached to dust particles could affect the survivability of airborne *E. coli*. In a study conducted by Zuo et al. [18], the authors mentioned that carrier particle size had a significant effect on the survivability of airborne viruses. Lighthart et al. [19] also reported that test bacterial survivability increased directly with droplet size. However, most of the studies used droplets as aerosol particles to carry bacteria and viruses. The dry dust particles may yield different results compared to droplets. So, the size distribution of airborne *E. coli* attached to dry dust particles also needed to be investigated.

This study aimed to investigate the survivability of airborne and settled *E. coli* via dry aerosolization under room thermal conditions. In addition, the survivability of *E. coli* in poultry litter was also investigated as a reference parameter.

## 2. Materials and Methods

To investigate the survivability of the airborne *E. coli* and the settled *E. coli*, experiments were run in a test chamber in a Biosafety Level 2 (BSL-2) laboratory. The survivability test of *E. coli* in poultry litter was conducted in Biosafety Level 1 (BSL-1) laboratory. Both laboratories are located at the Animal Science Department, University of Tennessee, Knoxville, TN 37996, USA.

### 2.1. Microorganism and System Descriptions

#### 2.1.1. Preparation of *E. coli* Solution

The *E. coli* strain used in this study was *Escherichia coli* GFP (ATCC^®^ 25922GFP™) which was purchased from American Type Culture Collection (ATCC, Manassas, VA, USA). *E. coli* strain was cultured at 37 °C, 150 rpm for 24 h in ATCC^®^ Medium 2855 (Tryptic Soy Broth ‘TSB’ with 100 mcg mL^−1^ Ampicillin and Tryptic Soy Agar ‘TSA’). The bacterial concentrations of *E. coli* in the solution after 24 h were from 8 to 9 log_10_ colony-forming units (log_10_ CFU) mL^−1^.

#### 2.1.2. Litter Preparation

Litter from the commercial broiler farm was first collected and stored in a container. It was then brought back to the BSL-1 laboratory to analyze the dry matter content. After that, the litter was autoclaved at 121 °C in 20 min and divided into identical-size aluminum boxes with the amount of 6 kg per box. The autoclaved poultry litter was used as a source of organic matter to simulate the biological conditions in poultry environment [20]. The sterilization was confirmed to demonstrate a state of freedom from microbial contamination. The boxes were sealed by aluminum foils and covered by plastic caps to avoid contamination. They were stored in a 4 °C fridge until being used.

It was important to prepare litter so that the bacteria were evenly distributed. To do that, 240 g of litter needed for the survivability test of airborne *E. coli* and settled *E. coli* experiment were equally distributed into 40 ceramic cups (6 g litter per cup). The amount of airborne dust that can be generated using a mixer was determined in a previous experiment [21], and the results showed that 240 g of litter would produce dust concentrations ranging from 0.9 to 1.1 mg m^−3^ which was within a typical range of dust concentration in commercial poultry farm [22]. To prepare litter inoculated with *E. coli*, litter in each of the 40 cups was mixed with 6 mL of *E. coli* solution. The 6 mL bacteria solution was sprayed evenly onto the litter in each cup. In the meantime, an aluminum spoon was used to gently mix the litter and *E. coli* solution. The mixtures then went through a process of drying at 22 °C and 52–67% relative humidity (RH) for 48 h until the dry matter content (DMC) of the mixture reached about 70%. The *E. coli* concentration in each cup was approximately 4 log_10_ CFU mg^−1^ litter after the drying process. The litter containing *E. coli* was then transferred from 40 ceramic cups to a metal bowl of the mixer for aerosolization. In the bowl, the litter was gently mixed up again before aerosolization.

#### 2.1.3. Test Chamber

Aerosolization was performed in an acrylic chamber. This chamber (2100 series, Cleatech, Orange, CA, USA) was a non-vacuum unit with two internal access doors with stainless steel frame, and a removable fully gasketed back wall. The dimension of the test chamber was 1.5 mL × 0.6 mW × 0.6 mH. The chamber was well sealed to prevent dust-laden particles from spilling out. It was also equipped with a temperature and RH sensor for continuously monitoring the inside thermal environment.

In the settled *E. coli* experiment, the chamber was modified to create a highly dusty environment in order to collect adequate settle dust for analysis. Initial results showed that the aerosolization space of the entire chamber was too large which led to the low concentration of airborne *E. coli* and dust particles. Thus, the chamber was modified by halving the aerosolization space using a partition acrylic film. The aerosolization space after modification was 0.75 mL × 0.6 mW × 0.6 mH.

#### 2.1.4. Aerosolization System

A stand mixer (model DCSM350GBRD02, New York, NY, USA) was used for dry aerosolization of airborne *E. coli* in this study. The dimension of the mixer was 0.3 mL × 0.2 mW × 0.3 mH with a 3.3 L stainless steel bowl. They operated at the highest speed to ensure the bacteria concentration in the air was high enough. A stir fan was also used to distribute the airborne *E. coli* in the chamber evenly.

#### 2.1.5. Dust Concentration Monitor

To monitor the dust concentration throughout the experiment, a dust concentration monitor (DustTrak DRX aerosol monitor 8533, TSI Inc., Shoreview, MN, USA) was used to provide data on the mass concentration of dust particles with different sizes. DustTrak was capable of measuring dust particles of PM 1, PM 2.5, PM 4.7, and PM 10. In this study, the dust concentration and particle size were recorded, and the results indicated that the particle concentration was relatively stable between experimental events.

#### 2.1.6. Air Samplers

To evaluate the survivability of the airborne *E. coli*, the AGI-30 impinger (AGI-30) was used to collect *E. coli*-laden dust particles in a test chamber in a BSL-2 laboratory. The AGI-30 operates at 12.5 L min^−1^. The airborne compounds were sucked through a fine nozzle in which the particles were accelerated and then impacted directly into the 20 mL TSB. The AGI-30 was proven to have the highest performance among three commonly used samplers (Andersen impactor, AGI-30 impinger, and BOBCAT ACD-200) for collecting airborne *E. coli* [21].

### 2.2. Experimental Design and Procedures

#### 2.2.1. Bacterial Size Distribution and Viable *E. coli* Recovering in the Airborne *E. coli* Survivability Test

An Andersen impactor was used to monitor the bacterial size distribution. The Andersen impactor is designed as an aerodynamic classifying system for airborne particles. It operates at 28.3 L min^−1^. Its six stages are designed to sort dust particles with different sizes of >7 µm, 4.7–7 µm, 3.3–4.7 µm, 2.1–3.3 µm, 1.1–2.1 µm, 0.65–1.1 µm, corresponding to stage 1 to stage 6. The dust particles carrying *E. coli*, after being aerosolized, were sucked in the intake on top of the Andersen impactor; then, the particles continuously went through 6 stages. For each stage, dust particles with sizes corresponding to each stage were collected on TSA agar plates.

In the process of sampling with the Andersen impactor, the stages of the sampler were often overloaded due to the excessive number of bacteria collected in each stage. Therefore, counting bacteria on agar plates directly was not possible. To overcome this problem, the agar plate washing method was applied. Bacteria, after being collected on agar plates, were immediately taken to the laboratory for analysis. Each agar plate was rinsed with 2 mL of TSB solution with the aid of a glass spreader, and then 1 mL of solution was collected by pipette. The 1 mL of this solution went through a traditional serial dilution process to determine the total *E. coli* in the solution. The agar plates, after washing, were also placed in an incubator letting the remaining *E. coli* on the plate grow. During the air sampling process, the agar plates in the Andersen impactor were dried by air flow in the sampler. Thus, the remaining 1 mL of solution in the washing process was mostly reabsorbed into the agar plates. However, to make sure that there is no residual solution that could affect the test results, the agar plates that have been washed instead of being turned upside down (due to traditional culture process) will be left right side up. The total *E. coli* on each stage was the combination of total *E. coli* collected from washing and total *E. coli* remaining on agar plates.

#### 2.2.2. Dry Matter Content Measurement

The moisture content is one variable affecting the survivability of bacteria [23]. The dry matter content (DMC), which is the inverse term of moisture content, was measured over time in the experiment. The DMC measurement of poultry litter is the ratio of the litter mass before and after the litter is completely dried. To determine DMC, the process was divided into two stages. First, the litter mass (m_1_) was weighted before going through a 48-h drying process until the litter mass was totally dried. After being dried at 105 °C, the litter mass (m_2_) was weighted again. The DMC was then calculated by the litter mass m_2_ divided by the litter mass m_1_.

#### 2.2.3. Sample Collection for Airborne *E. coli*

Two hundred and forty grams (240 g) of litter which contained ~4 log_10_ CFU mg^−1^ litter of *E. coli* were prepared and placed in the mixer. The mixer was placed in the center of the chamber to help evenly distribute the dust particles carrying *E. coli*. The mixer was fixed to the chamber surface by means of suckers, preventing it from moving during the running process. The stir fan was placed at the corner of the chamber to aid in distributing airborne particles. The AGI-30 was placed near the steel bowl of the mixer. 

Each test lasted a total of 50 min. The first 20 min of the test was the aerosolization process of airborne *E. coli* using the mixer and stir fan. After the 20-min aerosolization, airborne *E. coli* was collected using the AGI-30 for 10 min and the dust concentration was determined using DustTrak. The second sampling of airborne *E. coli* and dust followed the same protocol but was performed 10 min after the first sampling. This test procedure was repeated 7 times.

#### 2.2.4. Sample Collection for Settled *E. coli*

Two mixers were used for aerosolization. Two hundred and forty grams (240 g) of litter which contained about 4 log_10_ CFU mg^−1^ litter of *E. coli* were mixed gently and divided into two parts with 120 g for each mixer. The stir fan was operated during the aerosolization to improve the distribution of airborne *E. coli* in the chamber. Four Petri dishes were placed on both sides of the mixers to collect particles settled from the air. To avoid the position confounding effect, the Petri dishes were arranged randomly in a total of 4 experiment events. Each event started with 15 min aerosolization. After the aerosolization, the four Petri dishes were covered with caps and sealed by parafilm. Two Petri dishes were immediately analyzed to quantify viable *E. coli* via traditional culture technique. The remaining two Petri dishes were left at laboratory temperature at 20 °C, RH at 60% for 24 h. After that, they were quantified for viable *E. coli* by the same culture technique. The weight of each Petri dish was determined before and after aerosolization to determine the settle dust weight. The airborne dust concentration during the mixer running time was also monitored by DustTrak.

#### 2.2.5. Viable *E. coli* Counting for *E. coli* Survivability Test in Poultry Litter

Fifteen (15) ceramic cups, each with six grams (6 g) of poultry litter, were prepared to determine the survivability of *E. coli* in the litter. The six grams of poultry litter were spread in each ceramic cup so that the thickness of the litter was uniform and without large lumps. Then, 6 mL of *E. coli* solution was added to the litter by using a pipette. The solution was sprayed onto the litter, ensuring that the bacterial fluid was distributed as evenly as possible. After that, the mixture of litter and bacterial solution were mixed gently by using an aluminum spoon. The cup was then placed in the BSL-1 under laboratory conditions. The viable *E. coli* in the litter were determined at 0, 12, 24, 48, and 72 h after litter samples were prepared in the ceramic cups. At each time point, three cups of samples were used. In addition, two cups of litter added with the TSB solution instead of the bacteria solution were used as a control for *E. coli* analysis and DMC measurement.

To determine the viable *E. coli* counts, TSB was added in each cup so that the total volume of the mixture reached 15 mL. The mixture was mixed evenly. Then, 0.1 mL of the solution (litter-bacteria mixture mixed with TSB) was taken out and transferred to 0.9 mL of TSB. After that, the solution went through a serial dilution process to determine the counts of viable *E. coli*. By doing back-calculation, the bacterial concentration in poultry litter was calculated. 

#### 2.2.6. Determining *E. coli* Concentration in Poultry Litter

To determine the viable *E. coli*, the *E. coli* concentrations were calculated in logarithm colony-forming units per gram (log_10_ CFU mg^−1^) using Equation (1).
(1)$$C=\log_{10}\left(\frac{N\;\times 10^{n}}{V_{p}}\times {\;V}_{s}\times \frac{1}{m_{a}}\right),$$
where C = the bacteria concentration, log_10_ CFU mg^−1^; N = the number of colonies on a countable plate (30 to 300 colonies); n = serial dilution factor (n = 0 for undiluted sample, n = 1 for 10-fold diluted sample, etc.); V_P_ = the sample volume plated, mL (V_P_ = 0.1 mL in this study); V_s_ = the total volume of the original liquid sample, mL; m_a_ = the total poultry litter weight in each ceramic cup at the test time, mg.

#### 2.2.7. Determining Airborne *E. coli* Concentration

Each air sample collected by AGI-30 in liquid form (in TSB medium) was used to quantify viable *E. coli* via traditional culture techniques. After vortexing for 5 s, a 0.1 mL subsample, after going through the serially diluted (1:10) process, was plated onto TSA agar plates. In each experimental event, the subsample was uniformly repeated 3 times to ensure the accuracy of the experiment. The plates were aerobically incubated at 37 °C for 24 h. The visible *E. coli* colonies formed on plates (30 to 300 colonies) were determined. Based on the culture results and the sampled air volume, airborne *E. coli* concentrations were calculated in logarithm colony-forming units per cubic meter (log_10_ CFU m^−3^) using Equation (1). The parameter m_a_ converted to V_a_ which is the total air volume sampled using the bioaerosol samplers, m^3^.

#### 2.2.8. Determining Settled *E. coli* Concentration

Each settled sample on an empty Petri dish was used to quantify viable settled *E. coli*. After adding 10 mL of TSB medium (the culture medium) in each Petri dish, the Petri dish was gently shaken to wash the Petri dish surface and draw settled *E. coli* into TSB solution. After that, 0.1 mL of the solution containing *E. coli* was taken by using a pipette and went through a serial dilution process to count viable *E. coli*. Then, the viable *E. coli* was determined as the Equation (1). The parameter m_a_ was the mass of settled dust collected in each dish in each experiment, mg.

### 2.3. Calculation of Half-Life Time

The half-life time is the time interval needed for bacteria to decrease by half [24]. The bacterial concentrations throughout the experiments would be homogenized and normalized to the dust concentration (CFU mg^−1^). In the survivability of the airborne *E. coli* test, the airborne *E. coli* concentration was calculated based on airborne *E. coli* concentration collected in the air (CFU m^−3^) divided by total dust concentration (mg m^−3^). In the survivability of the settled *E. coli* test, the settled *E. coli* concentration was calculated based on the settled *E. coli* concentration collected on each Petri dish (CFU mg^−1^). The half-life time, then, was calculated by the following Equation (2).
(2)t1/2=log102× Tlog10(Cviable bacteriaC′viable bacteria),
where t_1/2_: half-life time (min or h); T = 20 (min) for airborne *E. coli* and 24 (h) for settled *E. coli* test; $C_{\mathrm{viable}\;\mathrm{bacteria}}$: *E. coli* concentration for the first sampling event, CFU mg^−1^; ${C^{\prime }}_{\mathrm{viable}\;\mathrm{bacteria}}$: *E. coli* concentration for the second sampling event, CFU mg^−1^. 

Linear simple regression was performed to calculate the half-life time of *E. coli*. The half-life time of *E. coli* in poultry litter was calculated based on the *E. coli* death over time by the linear Equation (3) [25]:(3)$$t_{1/2}=\frac{\mathrm{constant}\;-{\;\log}_{10}\left(\frac{C_{\mathrm{viable}\;\mathrm{bacteria}}}{2}\right)}{k},$$
where $C_{\mathrm{viable}\;\mathrm{bacteria}}:$ the *E. coli* concentration at 0 h, CFU mg^−1^; constant: intercept of the linear regression model, log_10_ CFU mg^−1^; k: the death rate, [log_10_ CFU mg^−1^] h^−1^; and $t_{1/2}$: half-life time, h. 

### 2.4. Statistical Analysis

Means and standard deviations for all experiments were calculated by using Rstudio (Rstudio, open-source license, Rstudio, Boston, MA, USA). Total 7 replicates for airborne *E. coli* experiment and 4 replicates for settled *E. coli* yielded decent statical analysis for calculating the half-life time. The conditions such as dust concentration among experiments were tested with the T-test to make sure there was no significant difference in terms of experimental conditions. The *t*-test significance level was 0.05 (*p* < 0.05). For the survivability of *E. coli* in poultry litter, at every time point, the concentration of *E. coli* in poultry litter was tested repeatedly 3 times for reliable viable *E. coli* data.

The half-life time of airborne *E. coli*, settled *E. coli* and *E. coli* in poultry litter were compared, and the differences between the survivability of *E. coli* under different conditions were tested by using a *t*-test run on Rstudio. The *t*-test was used to determine if the means of three sets of data (*E. coli* in poultry litter, airborne *E. coli*, and settled *E. coli*) are significantly different from each other. The *t*-test significance level was 0.05 (*p* < 0.05).

## 3. Results

### 3.1. Conditions for E. coli Survivability Test

Table 1 shows the litter DMC, initial litter *E. coli* concentration and environmental conditions during the experiments for determining survivability of airborne *E. coli*, settled *E. coli* and the *E. coli* in poultry litter. The DMC of litter, *E. coli* concentration and RH in the litter were kept stable throughout the experiments. In the test for settled *E. coli* survivability, instead of using one mixer, two mixers were used. Therefore, the heat generated in the two mixers caused the temperature in the test for settled *E. coli* survivability to be slightly higher than the two other tests.

### 3.2. Size Distribution of E. coli and Dust for the Airborne E. coli Survivability Test

The size distribution of airborne *E. coli* attached to dust particles and the size distribution of airborne dust particles were tested. The size distribution of airborne *E. coli* attached to dust particles during the 20-min aerosolization process is shown in Figure 1. The most *E. coli* were found in the particles larger than 7 µm with a percentage of 47.58%. The second large portion of *E. coli* was those attached to particles in the range of 4.7 to 7 µm, accounting for 27.34%. *E. coli* attached to dust particles in the ranges of 3.3–4.7 µm and 2.1–3.3 µm accounted for 14.05% and 9.92% of the total culturable *E. coli*, respectively. The least *E. coli* were found in particles smaller than 2.1 µm which accounted for 1.11% of the total culturable *E. coli*.

The size distribution of airborne dust particles during the 20-min aerosolization process was monitored by the DustTrak and shown in Table 2. Most dust particles have the size smaller than 1 µm with a concentration of 0.678 ± 0.108 mg m^−3^. The rest of the dust particles have size range of 1.0–2.5 µm, 2.5–4.7 µm, 4.7–10.0 µm and larger than 10.0 µm, with a concentration of 0.014 ± 0.001 mg m^−3^, 0.016 ± 0.005 mg m^−3^, 0.235 ± 0.042 mg m^−3^ and 0.232 ± 0.032 mg m^−3^, respectively. The total dust concentration was about 1.176 ± 0.120 mg m^−3^. As shown in Table 2 and Figure 1, although most dust particles were smaller than 1 µm, the size distribution of bacteria attached to dust particles was mainly larger than 2.1 µm, accounting for 98.89%. This indicates that when it comes to airborne *E. coli*, most are attached to dust particles with the size larger than 2.1 µm.

### 3.3. E. coli Survivability in Poultry Litter

The survivability of *E. coli* in poultry litter was determined in a 72-h test under laboratory conditions and delineated in Figure 2. The temperature and RH remained stable throughout the test at 20.5 ± 0.3 °C and 36 ± 4%. The DMC of litter (containing *E. coli*) changed throughout the test and was presented in Figure 2. The *E. coli* concentration decreased from 4.5 log_10_ CFU mg^−1^ to 2.4 log_10_ CFU mg^−1^ over 72 h. The DMC increased from 38% to 82% due to moisture evaporation. The half-life time of *E. coli* in poultry litter calculated based on the linear regression was 15.9 ± 1.3 h.

### 3.4. Airborne E. coli Survivability

The data collected from the first sampling and the second sampling to calculate the half-life time of *E. coli* were listed in the Table 3. As shown in Figure 1, most of the airborne *E. coli* were attached to dust particles larger than 2.1 µm, while only a small amount of total *E. coli* (1.11%) attached to dust particles smaller than 2.1 µm. Therefore, when calculating the concentration of *E. coli* in dust, we only considered the concentration of dust particles larger than 2.1 µm. The DustTrak was able to monitor the dust particles having size range of 1.0–2.5 µm, 2.5–4.7 µm, 4.7–10.0 µm and larger than 10.0 µm. In this study, we assumed that the amount of dust particles larger than 2.1 µm were equivalent to the amount of dust particles larger than 2.5 µm. The half-life time of the airborne *E. coli* based on dust with size > 2.5 µm was 5.7 ± 1.2 min.

### 3.5. Settled E. coli Survivability

The survivability of settled *E. coli* was tested over 24 h. In 24 h, the concentration of settled *E. coli* declined from 3.7 ± 0.1 to 3.0 ± 0.2 log_10_ CFU mg^−1^, yielding a half-life time of 9.6 ± 1.6 h for settled *E. coli*.

## 4. Discussion

The aim of this study was to determine the survivability of airborne and settled *E. coli* in laboratory under dry aerosolization conditions. Survivability of *E. coli* was determined using half-life time as the indicator. To calculate the half-life time, concentrations of airborne *E. coli* and settled *E. coli* collected at two different time points after the dry aerosolization process were measured and compared. The survivability of *E. coli* in poultry litter that was used for dry aerosolization was also determined in a 72-h test under laboratory conditions (20.5 ± 0.3 °C and 36 ± 4%). The results show that half-life times of airborne *E. coli*, settled *E. coli*, and *E. coli* in poultry litter were 5.7 ± 1.2 min, 9.6 ± 1.6 h, and 15.9 ± 1.3 h, respectively.

In the airborne *E. coli* survivability test, the mean half-life time of the bacteria based on dust particles with size larger than 2.5 µm was 5.7 min. Hoeksma et al. [26] tested survivability of airborne *E. coli* under wet aerosolization conditions at 20 °C and 40–60%. Their results showed that the half-life time of airborne *E. coli* under wet aerosolization conditions was about 2 min, which was much shorter than the half-life time calculated in the present study. The difference between the half-life time of airborne *E. coli* under wet aerosolization conditions and dry aerosolization conditions could be explained by inactivation due to evaporation. After being aerosolized, the wet aerosols lost their water film due to evaporation and become sensitive to ambient influences [26]. Moreover, the difference in preparation of *E. coli* for aerosolization between the two studies could be another reason of the discrepancy in survivability results. In the current study, the *E. coli* were prepared in poultry litter and exposed at laboratory conditions over 48 h before aerosolization. As such, the *E. coli* had already gone through a dehydration process before aerosolization, which might leave only dehydration-resistant *E. coli* for following dry aerosolization. In the study by Hoeksma et al. [26], the *E. coli* were aerosolized immediately after preparation via the wet aerosolization. In addition, the autoclaving process of poultry litter could affect the quality of poultry litter and produce Maillard reaction product. The Maillard reaction products were proven to inhibit growth of bacteria [27]. However, the effect of the preparation procedure was not well-studied in the present study. Therefore, the effect still needs further investigation.

Survivability and transmission range of airborne *E. coli* may be affected by the size of particles that *E. coli* attached to. Zuo et al. [18] reported that the carrier particle size had a significant influence in the transmission and survivability of airborne virus. In their study, the authors mentioned that the survivability of virus attached to larger particles was much longer than that attached to smaller particles. The possible explanation presented by Zuo et al. [18] was the shielding effect. In other words, compared with viruses existing as a singlet or attaching to small particles, the virus attached to larger particles could be better protected from changes of ambient environment [28]. The concentration of *E. coli* should be proportional to the weight of airborne dust in the entire size spectrum, assuming a uniform mixture of *E. coli* and poultry litter. However, most of dust particles were smaller than 1.0 µm (accounted for 57.60%) and the majority of airborne *E. coli* were found to attach to dust particles larger than 2.1 µm (98.89%). This contradiction could be explained again by the shielding effect. While *E. coli* attached to large particles could be protected from ambient influences, *E. coli* attached to small particles received less protection effect. It led to a rapid death of the *E. coli* attached to small particles during the aerosolization and sampling. 

The half-life time of settled *E. coli* in this study was about 9.6 h. Wilks et al. [16] tested the survivability of *E. coli* on metal surfaces at laboratory conditions at 20 °C. In their study, the total number of viable *E. coli* dropped by 1 log after the first 3 h, translating into an approximate 0.9 h half-life time. This discrepancy can be explained by differences in *E. coli* preparation methods, surfaces, and substrate (litter vs. liquid solution). As mentioned above, the *E. coli* preparation procedure in our study may affect the *E. coli* quality. Another possible explanation was metal surfaces used by Wilks et al. [16]. While the present study used regular plastic Petri dishes to collect settled *E. coli*, Wilks et al. [16] applied *E. coli* directly onto metal surfaces. This different material of surfaces could yield different survivability of *E. coli*. Ketkar et al. [29] indicated that stainless steel had antimicrobial effects. Further, different substrates (litter vs. liquid solution) used might have yielded different survivability of *E. coli*. While factors like pH and nutrient in poultry litter includes many affecting the survivability of bacteria [30,31], liquid solution used by Wilks et al. [16] for culturing *E. coli* was designed as a substrate for bacterial growth.

In the test of *E. coli* survivability in poultry litter, the half-life time was reported to be 15.9 ± 1.3 h. Compared with the half-life time of settled *E. coli* (9.6 h) and airborne *E. coli* (5.7 min), the half-life time of *E. coli* in poultry litter was significantly longer. A possible explanation was that the *E. coli* in the poultry litter did not go through the aerosolization process which negatively affect the *E. coli* survivability [32]. While settled *E. coli* and airborne *E. coli* were aerosolized, *E. coli* in the poultry litter were not aerosolized. In addition, degree of sample exposure to the environment could affect the survivability of *E. coli* as well. Ruiz et al. [33] reported that bacterial survival was highly influenced by ambient influences. The airborne *E. coli* were scattered in the air and the settled *E. coli* were prepared in thin layers where *E. coli* were exposed to ambient environment and were more susceptible to microenvironment changes [34,35], as compared to *E. coli* in the poultry litter. In contrast, the *E. coli* in poultry litter existed in a chuck form could be more protected from microenvironmental effects [34,35,36]. 

## 5. Conclusions

The study determined the survivability of airborne, settled, and poultry litter *E. coli* under dry aerosolization conditions in laboratory. Based on the results, we conclude that (1) most *E. coli* could be carried by the dust particles with aerodynamic diameter >2.1 µm, (2) the settled *E. coli* and the *E. coli* in poultry litter can survive much longer than airborne *E. coli*, and the mean half-life time was 5.7 ± 1.2 min for airborne *E. coli*, 9.6 ± 1.6 h for settled *E. coli*, and 15.9 ± 1.3 h for *E. coli* in poultry litter.

## Figures and Tables

**Figure 1 animals-12-00284-f001:**
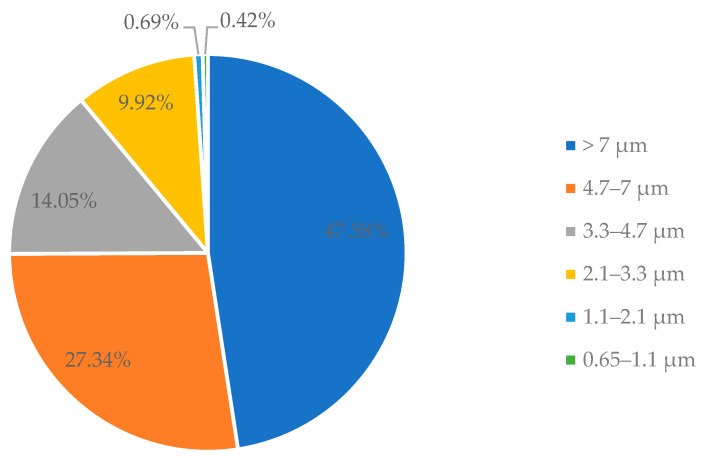
Size distribution of the airborne *E. coli* attached to dust particles in the airborne *E. coli* survivability test measured by an Andersen impactor.

**Figure 2 animals-12-00284-f002:**
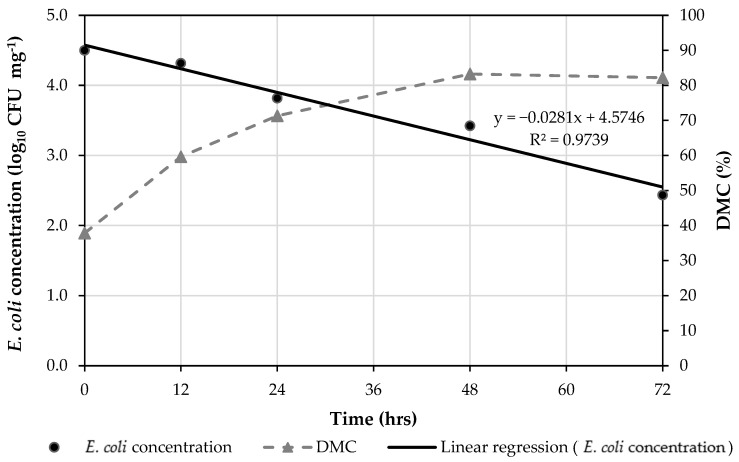
*E. coli* concentration and dry matter content (DMC) in poultry litter in a 72-h exposure under laboratory environmental condition (20.5 °C and 36%).

**Table 1 animals-12-00284-t001:** Conditions (Mean ± SD) for the *E. coli* in survivability test.

*E. coli* Concentration and Environmental Conditions	Test for Airborne *E. coli* Survivability	Test for Settled *E. coli* Survivability	Test for *E. coli* in Poultry Litter Survivability
DMC ^1^ of litter (%)	71 ± 5	72 ± 1	- ^2^
*E. coli* concentration in litter (log_10_ CFU mg^−1^)	4.4 ± 0.6	4.0 ± 0.5	- ^2^
Relative humidity (%)	54 ± 5	63 ± 7	36 ± 4
Temperature (°C)	22.1 ± 1.4	27.7 ± 5.1	20.5 ± 0.3

^1^ Dry matter content, ^2^ DMC and bacteria concentration varied over 72 h.

**Table 2 animals-12-00284-t002:** Dust size distribution (Means ± SD) in the airborne *E. coli* survivability test.

<1.0 µm (mg m^−3^)	1.0–2.5 µm (mg m^−3^)	2.5–4.7 µm (mg m^−3^)	4.7–10.0 µm (mg m^−3^)	>10.0 µm (mg m^−3^)	Total (mg m^−3^)
0.678 ± 0.108 (57.60%) ^1^	0.014 ± 0.001 (1.20%) ^1^	0.016 ± 0.005 (1.40%) ^1^	0.235 ± 0.042 (20.00%) ^1^	0.232 ± 0.032 (19.80%) ^1^	1.176 ± 0.120 (100.00%) ^1^

^1^ Percentage of the total for each size range.

**Table 3 animals-12-00284-t003:** Concentrations (Mean ± SD) of dust particles with size larger than 2.5 µm, airborne *E. coli* and airborne *E. coli*-to-dust ratio during air sampling for survivability test of airborne *E. coli*. The 2nd sampling was performed 20 min after 1st sampling.

Concentrations of Dust Particles and Airborne *E. coli*	1st Sampling	2nd Sampling
Dust concentration with size > 2.5 µm (mg m^−3^)	0.032 ± 0.022	0.016 ± 0.012
Airborne *E. coli* concentration (log_10_ CFU m^−3^)	7.1 ± 0.7	5.7 ± 1.0
Airborne *E. coli* concentration carried by dust concentration with size > 2.5 µm (log_10_ CFU mg^−1^)	8.7 ± 0.7	7.5 ± 0.9

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
