# Peer review of "Survival of Escherichia coli in Airborne and Settled Poultry Litter Particles"

_animals, 2022, doi:10.3390/ani12030284_

Round 1
Reviewer 1 Report
This manuscript aims to investigate the survivability of airborne and settled E.coil via dry aerosolization and the survivability of E.coil in poultry litter under room thermal conditions. The author found the half-time of airborne E.coil, E.coil in poultry litter, and settled particles. In my opinion, the biggest contribution of the authors of this manuscript is to explore the half-life of airborne E.coil by innovating a new method of dry aerosolization. This is a well-written paper containing interesting results which merit publication. For the benefit of the reader, however, a number of points need clarifying and certain statements require further justification:
Major comments:
1 Line 19-Line 38: Major parts of the abstract describe the background, objective, and methods of the experiment. There was a lack of description of the results and conclusions. For example, an important conclusion is “most E. coli could be carried by the dust particles with aerodynamic diameter > 2.1 µm”. However, this conclusion is not reflected in the abstract.
2 The content of Materials and methods were divided into too many subsections. It is suggested to reduce the subsections to the following sections: experiment platform, experiment design, concentration calculation equation, and Statistical analysis.
3 Line190-194: The author used 2 mL of TSB solution to rinse the agar plate, but collect 1 mL of the solution to go through a traditional serial dilution process to determine the total E.coil in the solution. Can the agar plates after washing absorb the residual solution? Because the plate needs to be turned upside down in an incubator during the culture process. Please describe this step in detail.
Minor comments:
1 Line 52: “ Enterobacteriaceae” change to “ Enterobacteriaceae”.
2 Line 58: What’s the meaning of “APEC”, please show the full name of the abbreviation.
2 Line 181, Line 323, Line 324, and Figure 1: The stage 2 of the Anderson impactor collects the dust particle with a size of “4.7-7.1 mm” instead of “4.7-7 mm”.
3 Line 278 and Line 288: Please change “formula” to “equation” as Lin 255 to unify the text.
4 Line 352: “E.coil concentration” change to “E.coil concentration”.
Reviewer 2 Report
The aim of the research was to determine the survivability of the E. coli - a common microbial species found in poultry environment - in airborne particles, settled dust, and poultry litter under laboratory environmental conditions. The number samples used in the experiment is sufficient. The applied research methods are correct. The Introduction chapter needs to be completed. The discussion is well conducted and comprehensive. Well-chosen references. References must be made in accordance with the instructions for authors. Before publishing in Animals, the article requires additions and corrections. The proposed changes are listed below:
General comments:
The References section must follow the instructions for authors. A template from the Animals journal manual is provided below
- Author 1, A.B.; Author 2, C.D. Title of the article. Abbreviated Journal Name Year, Volume, page range.
- Author 1, A.; Author 2, B. Title of the chapter. In Book Title, 2nd ed.; Editor 1, A., Editor 2, B., Eds.; Publisher: Publisher Location, Country, 2007; Volume 3, pp. 154–196.
- Author 1, A.; Author 2, B. Book Title, 3rd ed.; Publisher: Publisher Location, Country, 2008; pp. 154–196.
- Author 1, A.B.; Author 2, C. Title of Unpublished Work. Abbreviated Journal Name year, phrase indicating stage of publication (submitted; accepted; in press).
- Author 1, A.B. (University, City, State, Country); Author 2, C. (Institute, City, State, Country). Personal communication, 2012.
- Author 1, A.B.; Author 2, C.D.; Author 3, E.F. Title of Presentation. In Proceedings of the Name of the Conference, Location of Conference, Country, Date of Conference (Day Month Year).
- Author 1, A.B. Title of Thesis. Level of Thesis, Degree-Granting University, Location of University, Date of Completion.
- Title of Site. Available online: URL (accessed on Day Month Year).
For example, for original papers it must be Abbreviated name journal in italic, rok in bold, volume number in italic, page range using a long "ï€" from the Insert Symbol function
- When describing the meaning, use lowercase "p" in italics, spaces before and after "<" for example (p < 0.05)
Detailed comments:
In the INTRODUCTION chapter, write something about E. coli feed contamination; methods of reducing microbial contamination of the farm microclimate (chemical - disinfectants, formaldehyde, biological - effective microorganisms: e.g. Stęczny & Kokoszyński, 2021 in Animal Biotechnology; physical (UV rays); concentration of E.coli in airborne particles, settled dust, and poultry litter on commercial broiler chicken farms; microbiological methods to control the effectiveness of disinfection of the farm
L14 poultry litter? (straw, chaff, rye or wheat; wood shavings, maybe peat?) Write specifically what kind of litter
L23 also marked survivability of E. coli in the poultry litter, see L16, L32-34
L23 The aim of the research formulated in this way is confusing for the reader of this article
L42 „The United States of America (USA)” instead of The U.S.
L47 USA instead of U.S.
L316 + Please describe the data from Table 1
L322 "The most" instead of the „most”
L334 0.678 instead of 0.680 see Table 2
L431 [26,27] instead of [26; 27]
L443 [30,31] instead of [30; 31]
